# Is EDSS Enough to Predict Risk of Upper Urinary Tract Damage in Patients with Multiple Sclerosis?

**DOI:** 10.3390/biomedicines10123001

**Published:** 2022-11-22

**Authors:** Kevin Stritt, Ilaria Lucca, Beat Roth, Nuno Grilo

**Affiliations:** Department of Urology, Centre Hospitalier Universitaire Vaudois, University of Lausanne, 1011 Lausanne, Switzerland

**Keywords:** multiple sclerosis (MS), neurogenic lower urinary tract dysfunction (nLUTD), Expanded Disability Status Scale (EDSS), retrospective study, urodynamic study

## Abstract

Lower urinary tract dysfunction is often observed in patients with multiple sclerosis (MS) and may be responsible for an increased risk of upper urinary tract (UUT) damage. Although there are well-known urodynamic risk factors for UUT damage, no clinical prediction parameters are clearly identified. We aimed to confirm the accuracy of the Expanded Disability Status Scale (EDSS) in predicting urodynamic risk factors for UUT deterioration and to assess other clinical parameters potentially predicting urodynamic risk factors. We retrospectively reviewed 201 patients with MS referred for primary neuro-urological work-up, including a video-urodynamic study (VUDS) from August 2009 to February 2020. Multivariate modeling revealed EDSS, male gender, and a number of LUTS as clinical parameters significantly associated with urodynamic risk factors for UUT damage (*p* = 0.06, *p* = 0.01, *p* = 0.02, respectively). A nomogram combining EDSS, male gender, and a number of different LUTS was created to predict the presence of at least one urodynamic risk factor for UUT damage. In conclusion, the presence of high EDSS combined with male gender and several different LUTS is significantly associated with urodynamic risk factors and can be used to stratify MS patients for further neuro-urological assessment and treatment.

## 1. Introduction

Multiple sclerosis (MS) is the most common progressive neurological disorder in young people, with a mean age at onset of 30 years, and a prevalence of 133 cases per 100,000 people in Europe [1]. MS is a complex inflammatory and demyelinating disorder of the central nervous system [2]. The most common subtype, relapsing-remitting, accounts for approximately 80–85% of the cases at initial diagnosis [3]. Nearly 50% of these individuals develop a progressive course (secondary progressive) over a median time-period of 11 years [4]. Less commonly, patients might present a primary progressive course.

The urinary tract is often involved, and lower urinary tract symptoms (LUTS) are reported by >60% of patients with MS [5]. Symptoms might occur during the early stages of the neurological disease and sometimes might be reported at initial presentation. The urinary tract dysfunction and resultant LUTS present a psychosocial burden with a significant impact on quality of life and disability [6,7]. Moreover, aggravation of the underlying neurological disease may lead to the worsening of lower urinary tract dysfunction (LUTD), increasing the risk for urological complications such as upper urinary tract (UUT) deterioration, urinary tract infections (UTI) or kidney stones [6,8,9].

Urodynamic studies are frequently used in the management of MS-related LUTD and help in identifying patients at risk for UUT deterioration. Urodynamic risk factors that contribute to deterioration of the urinary tract include low bladder compliance, high maximum storage detrusor pressure, vesicoureteral reflux (VUR), and detrusor overactivity (DO) combined with detrusor sphincter dyssynergia (DSD), potentially leading to renal failure, increased mortality, and high overall costs [10,11,12].

Nonetheless, there is no consensus regarding the optimal urological management of patients with MS and existing guidelines are contradictory, especially considering referrals for video-urodynamic study (VUDS) [13,14,15,16].

It has been hypothesized that some clinical parameters can predict urodynamic risk factors for UUT deterioration [17,18]. A recent study of Ineichen et al. [17] found that the Expanded Disability Status Scale (EDSS) with a cutoff of 5.0 seems to be able to predict at least one urodynamic risk factor with a sensitivity of 86% and a specificity of 52%.

The aim of this study is to confirm the accuracy of EDSS in predicting urodynamic risk factors for UUT deterioration and to assess other clinical parameters potentially predicting urodynamic risk factors.

## 2. Materials and Methods

A retrospective analysis of medical records of patients with MS who underwent at least one VUDS for primary neuro-urological work-up at the department of Neurourology of Lausanne University Hospital since 2009 was performed.

The primary neuro-urological evaluation included medical history, clinical examination, urinalysis and urine culture, ultrasound of the urinary tract, and VUDS with pelvic floor electromyography. All the patients had at least one VUDS and more than half of them had two or more VUDS performed over time based on factors including urologic symptom changes and monitoring the success of bladder treatment. EDSS was collected to evaluate disability. The EDSS is an ordinal clinical rating scale ranging from 0 (normal neurologic examination) to 10 (death due to MS) in half-point increments [19]. The neurological disability was concomitantly assessed using the walking ability (ambulatory, cane, or wheelchair). MS was classified into the following three subtypes: relapsing-remitting, primary progressive, and secondary progressive. Treatments used for LUTD, including medical and surgical therapies, were recorded. The presence of LUTS, including urgency, frequency, urinary incontinence, and dysuria were recorded, as well as the number of urological complications, including recurrent urinary tract infections (rUTI), pyelonephritis, chronic kidney disease, and kidney stones. Patients with other neurologic conditions, or those with concurrent urologic malignancy, were excluded from the study.

The VUDS was conducted using a multichannel urodynamic system as recommended by the International Continence Society (ICS) [20]. Patients were examined in a sitting position and the bladder was filled with a mixture of 0.9% NaCl solution and contrast medium at room temperature at 30 mL/min. VUDS parameters were recorded in accordance with definitions established by international guidelines [21]. Normal bladder compliance was defined as little or no change in detrusor pressure during the process of the bladder filling (value of >40 mL/cmH_2_O). Detrusor overactivity (DO) was defined as involuntary detrusor contractions during the bladder filling that may be spontaneous or provoked. Detrusor sphincter dyssynergia (DSD) was defined as detrusor muscle contraction with concomitant and inappropriate involuntary urethral sphincter contraction during voiding. Detrusor underactivity (DU) was defined as low detrusor pressure or short detrusor contraction time, resulting in prolonged bladder emptying and/or a failure to achieve complete bladder emptying within a normal time span. Normal detrusor contraction during voiding ranges from 25 to 60 cmH_2_O. All studies were performed on a “Laborie Aquarius System^®^” (Laborie, Orangeburg, NY, USA) utilizing a “GE OEC^®^” c-arm (General Electric, Boston, MA, USA) by an experienced team and read by a dedicated urologist.

Risk factors for UUT deterioration were defined as generally agreed upon by the neuro-urology community, such as bladder compliance <20 mL/cmH_2_O [22], maximum storage detrusor pressure >40 cmH_2_O [23], VUR, and DO combined with DSD. The investigated variables were EDSS, age, age of MS onset, MS disease duration, sex, and number of LUTS.

The statistical analysis was performed using STATA 15.1 software (StataCorp LCC, College Station, TX, USA). A two-sided *p* < 0.05 was considered significant. Baseline variables were reported as either categorical or continuous. The Chi-square test was applied to compare categorical variables. Continuous variables that were normally distributed were reported as means and standard deviations, and differences between groups were examined using paired and unpaired Student’s *t*-test. Cox regression models were used to calculate the hazard ratios for multiple variables with a 95% confidence interval. Using the different predictors, a nomogram to predict the ability to detect at least one urodynamic risk factor was established.

## 3. Results

In total, 201 MS-diagnosed patients (139 females and 62 males) who underwent a VUDS for primary neuro-urological work-up were included. The mean (standard deviation (SD)) age of the patients was 51.5 years (11.5 years), the mean (SD) age of MS diagnosis was 34.9 years (10.3 years), and the mean (SD) duration of disease was 17.4 years (9.8 years). The relapsing-remitting (RR) MS subtype accounted for 61% of the cohort. The mean (SD) EDSS was 4.1 (2.1). Demographic and baseline characteristics are summarized in Table 1. All but four patients (98%) reported LUTS during the first visit with urgency (81%), urge incontinence (62%), frequency (53%), and dysuria (49%) being the most prevalent complaints. The mean (SD) number of different LUTS per patient was 2.6 (1.0). The mean (SD) daily pad usage was 1.7 (1.7). The most common VUDS pattern was DO + DSD (35%), followed by DSD alone (32%), and DO alone (12%). The normal VUDS pattern was present in 21% of cases. At initial VUDS, 15% of patients had already received medical treatment for LUTS (anticholinergic: 68%; alpha blockers: 32%). Clean intermittent catheterization (CIC) with or without concomitant medication was performed by 5% of the patients and 2% were already under botulinum toxin treatment.

The findings of VUDS from filling the cystometry and pressure flow studies are summarized in Table 2.

Generally, 118 (59%) of the 201 patients had at least one urodynamic risk factor for UUT deterioration, including bladder compliance <20 mL/cmH_2_O, maximum storage detrusor pressure >40 cmH_2_O, DO combined with DSD, or VUR. In total, 58 (29%), 27 (13%), and 2 (1%) patients presented 1, 2, and 3 urodynamic risk factors, respectively (Table 3).

A significant relationship was found between EDSS and the presence of at least one urodynamic risk factor for UUT deterioration (odds ratio = 1.33; 95% CI = 0.97–1.82; *p* = 0.08). Both male gender and a higher number of LUTS were associated with the presence of at least one urodynamic risk (odds ratio = 0.43; 95% CI = 0.22–0.85; *p* = 0.01 and odds ratio = 1.46; 95% CI = 1.06–1.99; *p* = 0.02, (Table 4)). No other clinical parameters were significantly associated with the presence of urodynamic risk factors for UUT damage.

When using the previously described EDSS cutoff of 5.0 to detect at least one urodynamic risk factor for UUT deterioration, a sensitivity of 40% and a specificity of 76% were achieved in the studied population. By reducing the EDSS cutoff to 4.0, the sensitivity increases to 60% and the specificity decreases to 69%. By combining EDSS ≥ 5.0 with a number of LUTS ≥ 3, the sensitivity rises to 78% but the specificity decreases to 27%. Therefore, a nomogram combining an EDSS ≥ 5, male gender, and the number of LUTS was created (Figure 1). 

## 4. Discussion

MS is a progressive and often debilitating disorder that may affect the urinary tract, causing storage and/or voiding dysfunction, which might increase the risk of UUT deterioration. At present, there is a lack of consensus across clinical guidelines regarding the optimal urological management of patients with MS, particularly considering referrals for urodynamic investigation. In this study, we evaluated clinical parameters potentially predictive of urodynamic risk factors for UUT damage. Risk determinants such as EDSS, male gender and a higher number of LUTS significantly increased the probability for patients to present at least one urodynamic risk factor. However, age of patients, age of MS diagnosis, and duration of MS disease, which were previously reported as possible risk factors [24], were not related to urodynamic risk factors in our cohort of patients after a multiple variable regression analysis.

In the study of Ineichen et al. [17], an EDSS cutoff of 5.0 detected almost 90% of the patients at risk for UUT damage with a false positive rate of <50%. This parameter alone does not provide a pragmatic, risk-dependent stratification by which to identify patients who require further neuro-urological assessment and treatment. Indeed, when applying this cutoff in our cohort, we detected a sensitivity of 40% and a specificity of 76% in predicting urodynamic risk factors. This difference might be mainly influenced by the fact that our population is composed of patients with a less severe neurological disease (mean EDSS (SD) 4.1 (2.1) vs. 5.1 (1.9)).

Moreover, risk determinants such as EDSS and male gender have already been described in the recommendations for neurologists and general practitioners. In these recommendations, in case of recurrent or complicated urinary tract infections, immunosuppressive treatments, EDSS ≥ 6 and male patients aged ≥ 55 years, referral to a neuro-urology unit with subsequent VUDS, bladder diary, and creatinine measurement should be provided [16]. A higher number of LUTS has not yet been studied as a risk factor, but it can help to identify patients requiring further neuro-urological assessment and treatment.

However, in patients with spinal cord injury or spina bifida, the risk for developing UUT damage and renal failure is much higher than in those with MS [5]. The burden of UUT damage and subsequent death among patients with MS is highly variable, ranging from none to 55% [25,26,27]. This variability is also highlighted in the different guidelines/recommendations. The main reason why there is no general agreement on the neuro-urological management of patients with MS is the lack of high-quality evidence studies, especially prospective studies assessing risk factors for UUT damage. 

The reliability of serial VUDS in patients with MS is not yet known. In addition, not all patients have significant neurogenic lower urinary tract dysfunction (nLUTD); thus, not all MS patients are susceptible to developing urinary complications. The 5th International Consultation on Incontinence and experts in the field do not recommend routine urodynamic investigations for the screening phase but suggest testing only when conservative measures fail [9,14]. Some other studies and reviews have not found a direct correlation between VUDS findings and patient-reported symptoms [28]. However, nLUTD is a major cause of morbidity and subsequent hospitalization in patients with MS [29]. Appropriate treatment can reduce the risk of UUT damage or even prevent it. Detecting patients at risk is essential to introducing appropriate treatment to protect renal function, improving quality of life, and decreasing disease-associated costs.

The idea of creating a nomogram combining EDSS, male gender, and number of LUTS seems an interesting way to stratify patients’ individual risk of significant urodynamic risk factors. Considering the findings of our retrospective study including 201 patients with MS, we recommend further neuro-urological assessment, including VUDS in patients with higher EDSS, male gender, and higher number of LUTS according to our nomogram. This can easily be incorporated into clinical practice, since EDSS and medical history are routinely obtained in patients with MS.

There are several limitations in this study. These include the observational retrospective nature of the study, the selection of the population and ongoing management.

## 5. Conclusions

In conclusion, patients with higher EDSS, male gender, and higher number of LUTS seem to be at a higher risk of presenting at least one urodynamic risk factor for UUT damage. We advise regular follow-up for this group of patients. In addition, screening MS patients for nLUTD based solely on an EDSS cutoff of 5.0 might lead to under-diagnosis and under-treatment. Combining EDSS, male gender, and number of symptoms, our nomogram can help guide clinical decision making. Other prospective studies are necessary to confirm this finding.

## Figures and Tables

**Figure 1 biomedicines-10-03001-f001:**
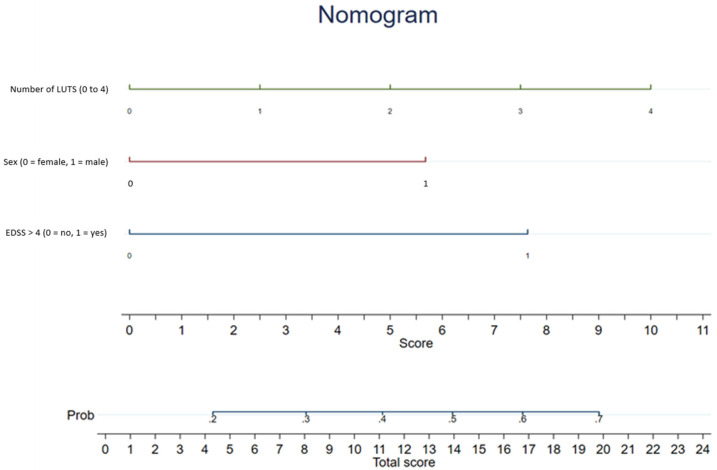
Nomogram combining an EDSS ≥ 5, male gender, and the number of LUTS to predict the probability of at least one urodynamic risk factor for UUT damage.

**Table 1 biomedicines-10-03001-t001:** Patients and disease characteristics.

Patients Characteristics	
Number of patients	201
Sex	
Female	139 (69%)
Male	62 (31%)
Age of patients (mean +/− SD)	51.5 +/− 11.5 years
Bladder emptying method	
Spontaneous voiding	183 (90%)
Clean intermittent catheterization (CIC)	9 (5%)
Suprapubic tube	4 (2%)
Indwelling urethral catheter	5 (3%)
Bladder medication	
None	170 (85%)
Anticholinergics	21 (10%)
Alpha blockers	10 (5%)
**Disease Characteristics**	
Age of MS diagnosis (mean +/− SD)	34.9 +/− 10.3 years
MS disease duration (mean +/− SD)	17.4 +/− 9.8 years
MS type	
Relapsing-remitting MS	124 (61%)
Secondary progressive MS	58 (29%)
Primary progressive MS	19 (10%)
Neurological disability	
Ambulatory	111 (55%)
Canes, crutches, and walkers	50 (25%)
Wheelchair bound	40 (20%)
EDSS (mean +/− SD)	4.1 +/− 2.1

MS: multiple sclerosis; EDSS: Expanded Disability Status Scale.

**Table 2 biomedicines-10-03001-t002:** Urodynamic parameters.

Filling Cystometry	
Maximum cystometric bladder capacity (mL)	371.0 +/− 168.5
Bladder compliance (mL/cmH_2_O)	99.7 +/− 56.4
DO	96 (48%)
Maximum storage detrusor pressure (cmH_2_O)	40.0 +/− 33.4
Reflex volume (mL)	234.2 +/− 162.0
DSD	136 (68%)
**Pressure Flow Study**	
Maximum flow rate (mL/s)	12.1 +/− 9.5
Maximum voiding detrusor pressure (cmH_2_O)	41.0 +/− 27.2
Detrusor pressure at maximum flow rate (cmH_2_O)	35.2 +/− 23.2
Voided volume (mL)	262.9 +/− 188.1
Post void residual (mL)	143.5 +/− 165.5

Mean +/− SD (range); DO: detrusor overactivity; DSD: detrusor sphincter dyssynergia.

**Table 3 biomedicines-10-03001-t003:** Urodynamic risk factors for upper urinary tract damage.

Bladder compliance <20 mL/cmH_2_O	*n* = 6
Maximum storage detrusor pressure >40 cmH_2_O	*n* = 40
VUR	*n* = 1
DO + DSD	*n* = 71

DO: detrusor overactivity; DSD: detrusor sphincter dyssynergia; VUR: vesicoureteral reflux.

**Table 4 biomedicines-10-03001-t004:** Univariable and multivariable logistic regression analysis prediction at least one urodynamic risk factor of UUT deterioration in patients with MS.

Variable		Univariable			Multivariable	
	OR	95% CI	*p*	OR	95% CI	*p*
EDSS	1.31	1.13–1.52	<0.001	1.33	0.97–1.82	0.06
Age	1.01	0.99–1.04	0.29			
Age of MS diagnosis	1.03	0.97–1.03	0.85			
MS disease duration	1.02	0.99–1.05	0.20			
Sex (male vs. female)	0.52	0.28–0.96	0.04	0.43	0.22–0.85	0.01
LUTS (number)	1.34	1.01–1.77	0.04	1.46	1.06–1.99	0.02

EDSS: Expanded Disability Status Scale; MS: multiple sclerosis.

## Data Availability

Not applicable.

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
