# Peer review of "Is EDSS Enough to Predict Risk of Upper Urinary Tract Damage in Patients with Multiple Sclerosis?"

_biomedicines, 2022, doi:10.3390/biomedicines10123001_

Round 1
Reviewer 1 Report
This article is very well written and is about the potential predictors of upper urinary tract (UTT) damage in MS patients. UTT damage is common in MS patients but is understudied. In their study which was retrograde, so had the limitation all retrograde studies, EDSS, male gender, and number of 15 LUTS seemed to be potential predictors of UUT damage.
PROS:
Well written and essential in terms of helping clinicians to know when to have a urology consult and VUDS for their MS patients as lower urinary tract problems are also very common and sometimes manageable by neurologist without urology referral.
CONS:
- The UUT damage that are regarded in this study did not cover kidney function, like GFR or urethral damage like hydronephrosis.
- Upper urinary tract function of patients with multiple sclerosis Violaine Piquet,Nicolas Turmel,Camille Chesnel,Rebecca Haddad,Frédérique Lebreton,Gérard Amarenco,Claire Hentzen, neurourology urodynamic 2021
- The mean age of patients in this study was 51.5 and while male sex is introduced as a predictor of UTT damage, it would have been good if they adjust for the prostate size and see if the effect remains consistent.
- There was no stratification of risk factors based on MS status of being still relapsing remitting or progressive.
- Possible evaluation of cord lesions and development of UTT might also be of interest.
Reviewer 2 Report
Dear Authors,
Manuscript entitled: "Is EDSS enough to predict risk of upper urinary tract damage in patients with multiple sclerosis?" is an interesting research article that could be published after minor changes. There are issues that require correction:
- all abbreviations should be explained, e.g. in the Abstract
- normal ranges for non-urologists should be added
- English language should be corrected
- Figure 1 should be made more clear for readers.
